# Construction of a Character Dataset for Historical Uchen Tibetan Documents under Low-Resource Conditions

**Ce Zhang [1,2]**, **Weilan Wang [1,*]** and **Guowei Zhang [3]**

1 Key Laboratory of China's Ethnic Languages and Information Technology of Ministry of Education, Northwest Minzu University, Lanzhou 730030, China
2 School of Artificial Intelligence, Chongqing University of Education, Chongqing 400065, China
3 LinkDoc Technology, Beijing 100089, China
* Correspondence: wangweilan@xbmu.edu.cn

**Abstract:** The construction of a character dataset is an important part of the research on document analysis and recognition of historical Tibetan documents. The results of character segmentation research in the previous stage are presented by coloring the characters with different color values. On this basis, the characters are annotated, and the character images corresponding to the annotation are extracted to construct a character dataset. The construction of a character dataset is carried out as follows: (1) text annotation of segmented characters is performed; (2) the character image is extracted from the character block based on the real position information; (3) according to the class of annotated text, the extracted character images are classified to construct a preliminary character dataset; (4) data augmentation is used to solve the imbalance of classes and samples in the preliminary dataset; (5) research on character recognition based on the constructed dataset is performed. The experimental results show that under low-resource conditions, this paper solves the challenges in the construction of a historical Uchen Tibetan document character dataset and constructs a 610-class character dataset. This dataset lays the foundation for the character recognition of historical Tibetan documents and provides a reference for the construction of relevant document datasets.

**Keywords:** historical Tibetan documents; character annotation; character extraction; data augmentation; character recognition

## 1. Introduction

The content recorded in historical Tibetan documents is extremely rich, involving all aspects of social development, and has important reference value for the study of Chinese culture. After much reading and preservation, historical Tibetan documents have been damaged to varying degrees, and some have even been lost. The protection, development and utilization of existing historical Tibetan documents have become an important and urgent topic in ethnic language research. Tibetan is a low-resource language, which makes it difficult to obtain a large amount of document data, and historical Tibetan documents are more difficult to obtain, resulting in the late start of relevant research. The study of historical Tibetan documents began in the 1980s. Since 1991, Kojima et al. have studied the analysis and recognition research on woodcut Tibetan documents, including character recognition [1,2], feature extraction [3] and other works. However, these works did not involve the construction of character datasets. More than a decade later, some researchers conducted relevant research in layout analysis [4], text line segmentation [5], character segmentation [6,7] on different versions of historical Tibetan document data. Since 2017, our research group has conducted more analysis and recognition research on historical Tibetan document images. Han et al. proposed a binarization approach based on several image processing steps, which achieved high performance in image binarization [8]. Zhao et al. proposed an attention U-Net-based binarization approach for the historical Tibetan document images [9]. Zhou et al., Wang et al. and Hu et al.

proposed a text line segmentation method based on contour curve tracking [10], based on the connected component analysis [11] and combined local baseline and connected component [12] for Tibetan historical documents, respectively. In addition, in the aspect of layout analysis, Zhao et al. proposed accurate fine-grained layout analysis for the historical Tibetan document based on instance segmentation [13]. Zhang et al. studied character segmentation on the basis of previous work, which provides data for the construction of a character dataset in this work [14].

Dataset construction is an important part of document analysis and recognition of historical Tibetan documents. In 2017, Wang et al. proposed an online handwritten "Tibetan-Sanskrit" sample generation method based on component combination [15]. First, the "Tibetan-Sanskrit" character set and component set were determined. Second, the location information of the component was obtained. Third, the samples of the online handwritten "Tibetan-Sanskrit" sample components were collected, and finally, a sample database of the "Tibetan-Sanskrit" character dataset was generated. In 2018, Li et al. constructed a dataset for the text recognition of historical Uchen Tibetan documents [16]. The samples of the dataset were composed of the upper part of the baseline, the lower part of the baseline and punctuation symbols. The number of samples in different classes was uneven, and there were no real samples corresponding to historical Tibetan documents in some classes. Because the character segmentation of historical ancient Tibetan documents is faced with the following challenges: text lines have different degrees of tilt and twist, there are many overlapping, crossing, touching and breaking character strokes; there are certain differences in the writing styles of the same characters in different positions. Up to now, there has been no relevant report on the character dataset with complete characters of historical Uchen Tibetan documents. Therefore, the construction of character datasets still faces two challenges.

Challenge 1: Historical Tibetan documents belong to the handwriting category. There are great differences in writing styles, resulting in large left-right deviations in character strokes. It is difficult to extract characters according to their real position information.

Challenge 2: Tibetan is a low-resource language, which makes it difficult to obtain a large amount of document data, and there are a large number of characters with low frequency, which aggravates the obvious difference in the numbers of samples of character classes.

In view of the above challenges in the construction of character datasets, we propose a character extraction and dataset augmentation method.

In short, the main contribution of this work is that we constructed a historical Uchen Tibetan document character dataset that has 610 classes, basically covering high-frequency characters. It lays the foundation for character recognition and provides a reference for the construction of historical document character datasets in other languages.

## 2. Related Work

Under low-resource conditions, the following related work has been performed for the construction of historical Tibetan document character datasets. The research results of the previous stage as the basis of this work (Figure 1) as follows: An example image of historical Tibetan documents shown in Figure 1a is from the Buddhist Digital Resource Center (BDRC) [17]. Character blocks after segmentation shown in Figure 1b,c are from our previous research results [14]. the characters in the segmented character block are annotated, extracted and augmented, respectively, and the character dataset is constructed. Finally, the character recognition experiment is conducted with the constructed character dataset, and the quality of the constructed character dataset has been verified. The research content framework of this paper is shown in Figure 2.

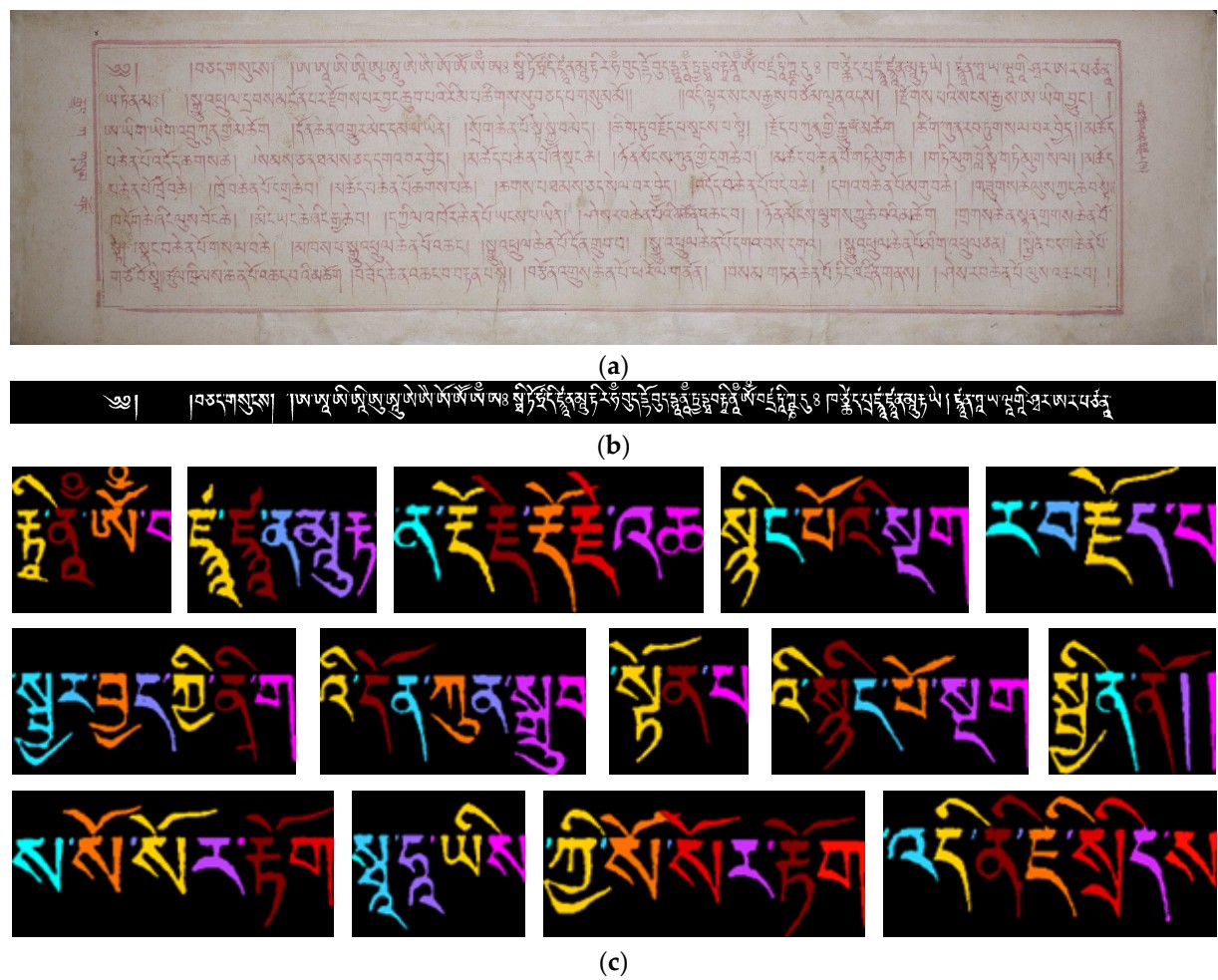

**Figure 1.** The research basis of this work: (**a**) example of historical Uchen Tibetan document; (**b**) example of text line after binarization and extraction; (**c**) partial character block obtained after a series of processes.

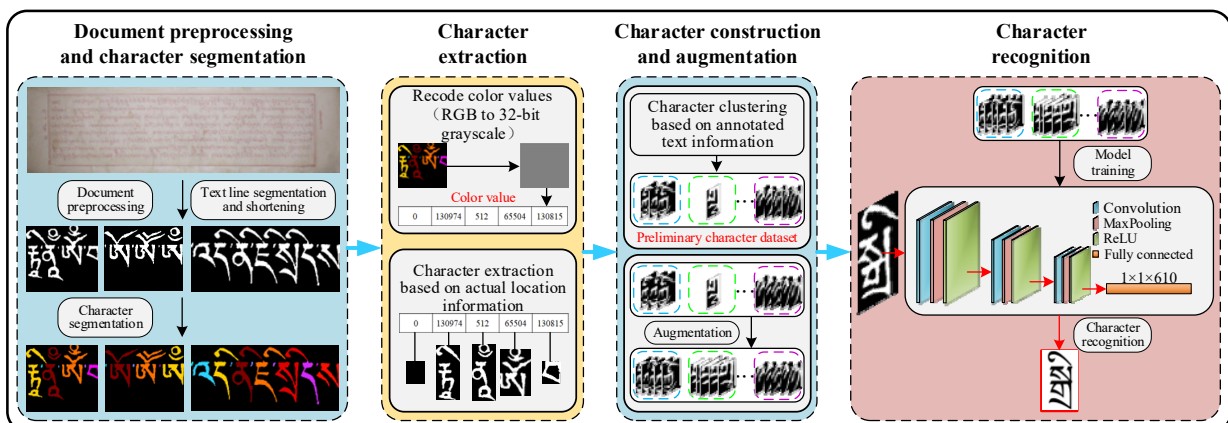

**Figure 2.** Construction framework of the character dataset of historical Tibetan documents.

### 2.1. Character Annotation

Character annotation is to annotate the character image with the corresponding encoded character text. Due to the reduction in the quality of historical Uchen Tibetan documents to varying degrees, some documents have become blurred, and there is much

Sanskrit Tibetan in historical documents, which brings multiple difficulties to character annotation. To improve the accuracy of annotation, we annotate character blocks.

*2.2. Character Image Extraction*

We directly use the research results of character segmentation to construct the character dataset. The characters in the character segmentation results are on character blocks with different widths, and different characters are colored with different RGB color values. In order to achieve one-to-one correspondence with the annotated text, the characters in the character block need to be extracted according to the real position order. It is an easy task to extract the colored characters by color values, but the strokes of historical Tibetan characters have different degrees of left and right offsets, and the shape differences between strokes are obvious. Directly using the centroid coordinates or color values of characters (coloring is not in a fixed order) may lead to inconsistency between the extracted character sequence and the real position sequence.

*2.3. Dataset Construction and Augmentation*

The annotated character text is classified, and the extracted character image is put into the corresponding class. Because Tibetan is a low-resource language and there are many low-frequency characters, the class and sample number of the character dataset are uneven. To solve the data imbalance problem, we use synthetic data to realize data augmentation.

*2.4. Character Recognition*

The constructed character dataset provides data for recognition. The quality of the dataset can be tested through character recognition. The constructed character dataset is used as the training set to train the recognition network. The correct character class is used as the test set to obtain the accuracy of character recognition.

**3. Methodology**

*3.1. Character Annotation*

Character image annotation is a heavy task. In view of the close relationship between the front and rear characters in Tibetan text, we first annotate in the character block before character segmentation and then realize character annotation through the corresponding relationship between the character image and character text in the character block. To ensure the accuracy of character annotation, we choose 8 Tibetans whose mother tongue is Tibetan and who have completed their graduate education as annotators to complete the annotation. Their annotation tasks do not overlap. An expert in the field of Tibetan language studies proofreads their annotation results.

Tibetan adopts the Unicode coding scheme to encode Tibetan basic characters. Tibetan is formed by superimposing basic characters up and down, and the letters of the characters where the base consonant is located can be superimposed up to 4 layers (Figure 3), including prefixed consonant (PFC), base consonant (BC), superscript consonant (SPC), subscript consonant (SBC), top vowel (TV), bottom vowel (BV), suffixed consonant (SFC), further suffixed consonant (FSFC), and at most 1 vowel (top vowel or bottom vowel) appears. Syllables are separated by syllable point (SP). There is an implied base line (BL) between the top vowel and the other letters. There is a large amount of Sanskrit Tibetan in historical Tibetan documents. Sanskrit Tibetan has only top-down superposition, up to 7 layers at most. In the Unicode encoding scheme of Tibetan [18], the encoding length of the same character is inconsistent (Table 1). Therefore, the annotated character text needs to be processed with a unified coding method, that is, the way that all annotated text is unified into the least number of coding units.

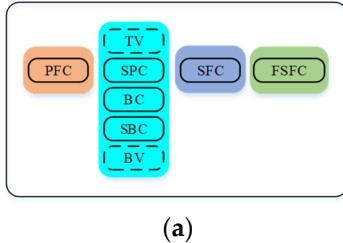 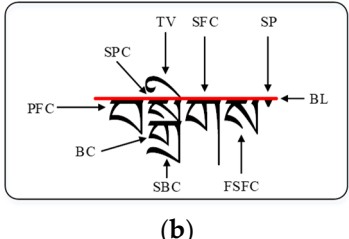

(**a**)　　　　　　　　　　　　　　　　　　　（**b**）

**Figure 3.** Syllable structure in Tibetan: (**a**) schematic diagram of syllable structure in Tibetan; (**b**) example of syllable structure in Tibetan.

**Table 1.** Corresponding examples Tibetan characters and Unicode coding.

| No. | Tibetan Character | Component Unit and Code | Character Code |
|---|---|---|---|
| 1 | ཨྰོཾ | | 0F00 |
| | | | 0F68 0F7C 0F7E |
| 2 | བྲྀཽབ | | 0F57 0FB2 0F75 0F83 |
| | | | 0F56 0FB7 0FB2 0F75 0F83 |

　　We extract the segmented characters according to the order of the character text in the annotated character (Figure 4). The segmented character blocks are formed by coloring in turn in the unit of characters; that is, all strokes of the same character are colored into one color. Before extracting characters from the character block after character segmentation, to avoid the touching or loss of syllable points in the character block, which would cause the characters in the annotated text to not correctly correspond to the characters in the character block, it is necessary to delete the syllable points in the annotated text and the character block to obtain the annotated text and character block after deleting the syllable points (Figure 4d,e).

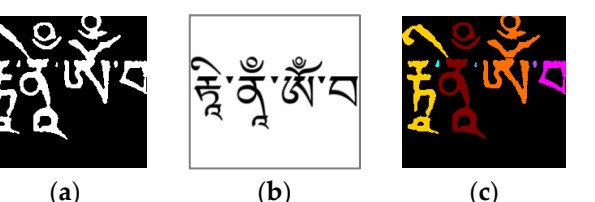 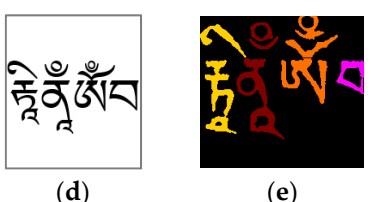

(**a**)　　　　　　(**b**)　　　　　　(**c**)　　　　　　(**d**)　　　　　　(**e**)

**Figure 4.** Example of character annotation: (**a**) character block; (**b**) annotated text; (**c**) character block after segmentation; (**d**) annotation after shielding syllable points; (**e**) character block after shielding syllable points.

### 3.2. Character Image Extraction

　　The characters of historical Uchen Tibetan documents belong are handwritten, and their stroke positions and shapes are quite different. In addition, the interference of three channel images in character extraction makes it difficult to extract the characters in sequence by using the centroid or coordinate position information of the connected component, which easily leads to disorder between the extracted character numbering sequence and the sequence. To solve the above problems, we first realize color value conversion by performing secondary encoding on the color values of the RGB three channels into a 32-bit grayscale and then extract the single characters, which can avoid the possibility that the

color values cannot be uniquely corresponding when the RGB image is converted to the gray-scale image.

We extract characters from encoded character blocks based on real location information (Algorithm 1).

---

**Algorithm 1:** Character extraction algorithm based on real location information

---

Input: Encoded character block ECB
Output: Single character SC
1: H, W ← size function(ECB)
2: NC ← count the number of colors in ECB
3: **for** $l$ = 1: 2: H **do**
4:　　ECBL ← ECB(1: $l$,:) = 0, ECB($l$ + 2, :) = 0
5:　　NCL ← count the number of colors in ECBL
6:　　**if** NC == NCL
7:　　**for** $c$ = 1: NC **do**
8:　　　**for** $h, w$ = 1: H, 1: W **do**
9:　　　　**if** ECB($h, w$) ≠ NCL($c$)
10:　　　　ECB($h, w$) = 0
11:　　　　**end if**
12:　　　**end for**
13:　　　SC ← cropping function(ECB)
14:　　**end for**
15:　　**end if**
16: **end for**

---

### 3.3. Dataset Construction and Augmentation

### 3.3.1. Character Dataset Construction

The classes of characters in the annotated text are counted to obtain the number of text classes. The annotated text string is extracted one by one according to the real order and named with the real order number, which lays the foundation for its subsequent matching with the character image. According to the extracted character number, the character image is classified into the corresponding character class to form a preliminary character dataset (Algorithm 2). There may be some errors in the process of character annotation and character extraction, and the character images in the character set need to be proofread manually.

---

**Algorithm 2:** Construction of the algorithm of the character dataset based on real position sequence matching

---

**Input:** Single character annotations SCA, Single characters SC
**Output:** Preliminary character dataset PCD
1: NSCA ← count the number of SCA
2: **for** $d$ = 1: NSCA **do**
3:　　MN ← read the mapping name of SCA[$d$]
4:　　ISEF ← exist the class folder of SCA[$d$] or not
5:　　**if** not ISEF
6:　　　CF ← create the class folder of SCA[$d$]
7:　　　the MN in SC is written into CF //the MN corresponds to a single character name
8:　　**else**
9:　　　the MN in SC is written into CF
10:　　**end if**
11:　　PCD ← save function(CF) //each CF represents a character class
12: **end for**

---

### 3.3.2. Character Dataset Augmentation

The character dataset established by segmented characters has obvious differences in the number of samples of classes. There are more commonly used characters in Tibetan, less

commonly used characters, and, even, no-character image samples in classes. We augment the character classes and samples in the character set by artificially synthesizing characters to balance the numbers of samples of character classes.

- Ordinary character
  We use characters synthesized by "Tibetan-Sanskrit" handwritten sample generation method based on component combination [15].

- Special symbols
  Because the component samples of special symbols are not collected in the "Tibetan-Sanskrit" handwritten sample generation method based on component combination, special symbols cannot be synthesized by this method. In addition to the syllable point ("·") and single vertical line ("|"), there are seven main kinds of special symbols commonly used in the document images of historical Uchen Tibetan documents, namely, "ༀ", "ༀ", "ༀ", "ༀ", "|", "|" and "ༀ". For the first four special characters mentioned above, we collect them directly from the images of historical Uchen Tibetan documents. The fifth to sixth special symbols are highly similar to the single vertical line in shape and structure. By cutting a point and two points on the head position above the single vertical line symbol, we can obtain the fifth and sixth symbols, respectively. For the seventh special symbol, due to the small number of historical Tibetan documents, the direct acquisition method has difficulty achieving the purpose of augmentation. Through observation, the upper and lower strokes of the seventh special symbol ("ༀ") have symmetry. The difference between the special symbols mainly lies in the spacing between the upper and lower strokes. We propose a special character synthesis method based on the random distance between upper and lower strokes (Algorithm 3).

---

**Algorithm 3:** Special character synthesis algorithm based on the random distance between upper and lower strokes

---

Input: Special symbols SS
Output: Synthesized special symbols SSS
1: NSS ← count the number of SS
2: **for** $s$ = 1: NSS **do**
3:    SSU, SSL ← extract the upper and lower strokes of SS //use the Y coordinate of centroid
4:    SSUs, SSLs ← rotate function(SSU, SSL) //rotate 90 degrees counterclockwise
5:    RI ← generate a random integer in [−2,12] //[−2,12] is the priori distance range
6:    SSL, SSU ← SSUs, SSLs //the upper(lower) stroke is as the lower(upper) stroke of SSS
7:    SSS ← synthesis function(SSL, SSU, RI)
8: **end for**

---

### 3.4. Character Recognition

There are many research achievements in Tibetan recognition. From the level of technology and method applied, it can be roughly divided into methods based on digital image processing technology and deep learning methods. The former mainly focuses on the design and extraction of character features [19,20] and the design of classification algorithms [21,22]. The latter mainly focuses on the design and training of recognition models [23–25]. The main purpose of character recognition in this paper is to verify the quality of the constructed character dataset. Considering that all the samples in the character dataset are single characters, the classical convolutional neural network LeNet-5 [26] is used in the character recognition model (Figure 5). The convolutional neural network has three convolution layers and one fully connected layer. See the following for the specific configuration. The character dataset comes from the constructed historical Uchen Tibetan character dataset, which basically covers 610 classes of the most commonly used characters and symbols in historical Uchen Tibetan documents, with 700 image samples in each class. The character dataset is divided into a training subset and a verification subset at a ratio of 8:2, and the test data come from the correct characters obtained by character segmentation.

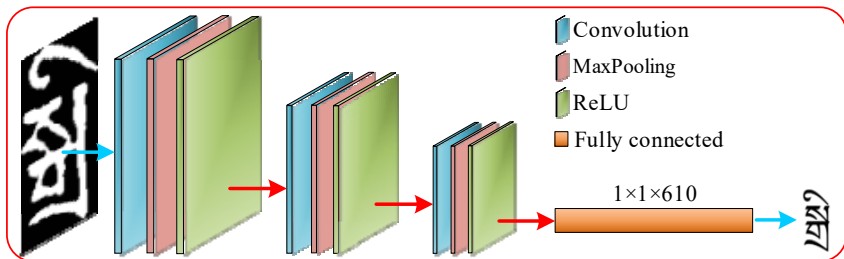

**Figure 5.** Character recognition model framework.

Historical Tibetan documents belong to the category of handwritten text. There are differences in writing styles among characters, and the stroke deformation of some characters is more serious. To objectively evaluate the advantages and disadvantages of this method, accuracy is used to evaluate the character recognition results, expressed as follows:

$$Accuracy = \frac{NCRC}{NRC} \times 100\% \tag{1}$$

where *NCRC* is the number of correctly recognized characters and *NRC* is the number of recognized characters.

## 4. Experimental Results and Analysis

Using the character blocks after character segmentation as the basic data, we carried out relevant research on character block annotation, character image extraction, character dataset construction and augmentation, and character recognition.

### 4.1. Character Image Extraction Results

Using Algorithm 1, single characters are extracted one by one from the correctly segmented character blocks (Figure 6).

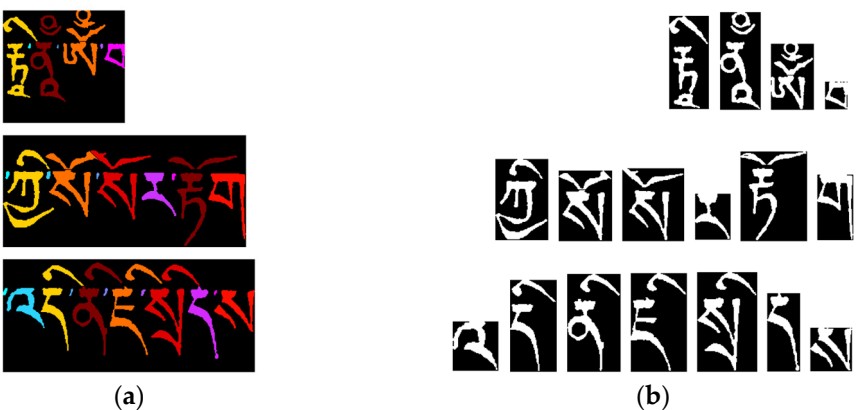

(**a**)             (**b**)

**Figure 6.** Example of character image extraction results after shielding syllable points: (**a**) character block after segmentation; (**b**) extracted characters (shielded syllable points).

### 4.2. Character Dataset Augmentation and Construction Results

For 610 classes of Tibetan characters, 700 images are selected as class samples. In addition, we synthesized 47 classes of characters missing from the character classes of Tibetan documents (Figure 7) and augmented classes with fewer than 700 samples in the existing classes (Figure 8).

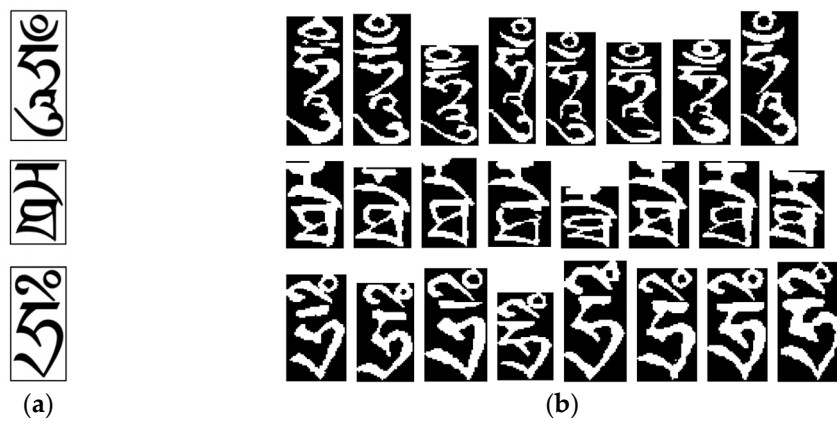

**Figure 7.** Example of synthesized character samples: (**a**) character text; (**b**) synthesized character image samples.

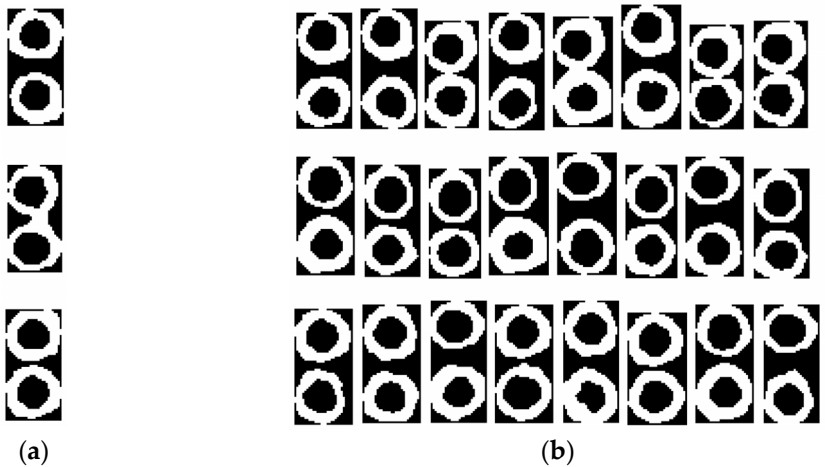

**Figure 8.** Example of character samples before and after augmentation: (**a**) character sample before augmentation; (**b**) character sample after augmentation.

After the above work, the character dataset of the historical Uchen Tibetan document and the corresponding annotation mapping file are finally constructed (Figure 9).

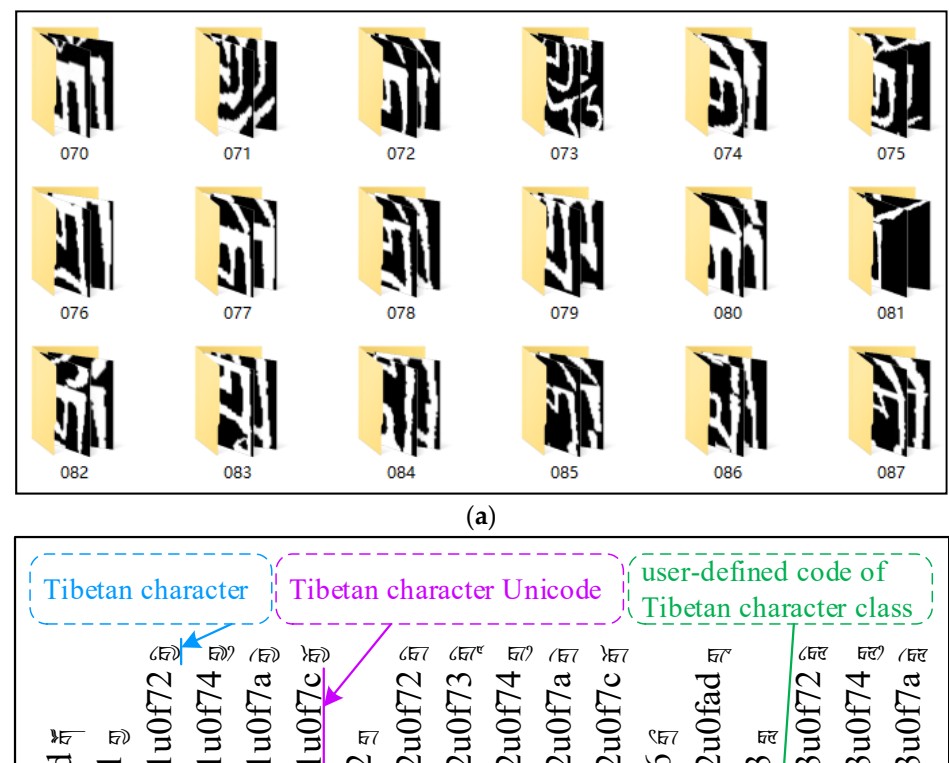

**Figure 9.** The constructed character dataset: (**a**) character images; (**b**) part of the character annotation mapping file (rotated 90 degrees to the right).

*4.3. Character Annotation and Recognition Results*

Character recognition is not only the purpose of character segmentation but also an important means of testing the effect of character segmentation. By dividing the established character dataset into a training set and a verification set at a ratio of 8:2, the model of the designed convolutional neural network is trained (the configuration is shown in Table 2). Finally, the trained recognition model is used to test the effect of character segmentation. The data to be recognized come from six sets of character block data obtained based on projection shortening in our character segmentation research results. The data to be recognized come from six sets of character block data obtained based on projection shortening in our character segmentation research results [14] in the previous stage. The ratio of the average height to the average width of characters in historical Tibetan documents is nearly 2:1, so the size of the input image is set to 64 × 32. The average loss value after each training and the recognition effect on the verification set are shown in Figures 10 and 11, respectively.

**Table 2.** Convolutional neural network configuration.

| Type | Configuration |
|---|---|
| Convolution | k5 × 5 |
| MaxPooling | k2 × 2, s2 |
| Convolution | k5 × 5 |
| MaxPooling | k2 × 2, s2 |
| Convolution | k5 × 5 |
| Linear | 1 × 1 × 610 |
| Softmax | 1 × 1 × 610 |

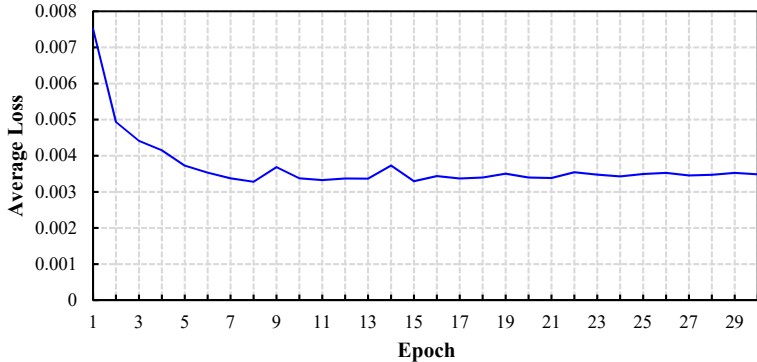

**Figure 10.** Loss value of the model training process.

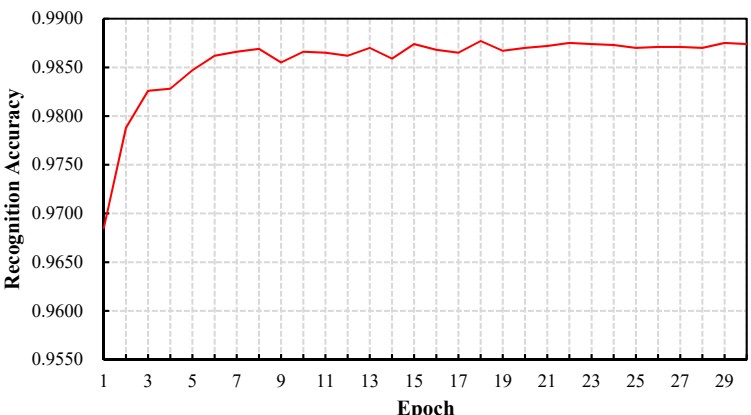

**Figure 11.** Recognition effect on the validation set.

The framework used for model training is Pytorch, and the computing platform is the CPU (Core i7-9700 3.00 GHz). The network model is trained using the CrossEntropyLoss loss function and the SGD optimizer (learning rate = 0.001, momentum = 0.9). The batch size and training epoch are set to 64 and 30, respectively.

Syllable point touching is a common phenomenon in historical Tibetan documents. Due to the small area of the syllable point, the characters formed after the syllable point touching show little change in character structure attributes. Without the support of semantic knowledge, it is extremely difficult to determine the touching of the syllable point. Therefore, touching the syllable point and characters is not discussed in this paper. To ensure that the characters in the segmented character block can correspond to their annotated text in turn, we shield the non-touching syllable points in the character block and the syllable points corresponding to the annotated text.

The data to be recognized comes from six sets of correctly segmented character blocks, which were obtained through systematic sampling and random sampling (Table 3). The three stroke attribution methods in Table 3 come from the centroid-based attribution method

(Cen-AM), the connected component-based attribution method (Con-AM) and the attribution method combining the connected component and centroid methods (ConCen-AM) in our character segmentation work [14] in the previous stage, where NCCB is the number of correct character blocks, NACB is the number of annotated character blocks, PACB is the proportion of annotated character blocks, and NCASS is the number of characters after shielding syllables.

**Table 3.** Annotating the character block to be recognized.

| Sampling Method | Stroke Attribution Method | Data Abbreviation | NCCB | NACB | PACB | NCASS |
|---|---|---|---|---|---|---|
| Systematic sampling | Cen-AM | SS-Cen | 9916 | 5581 | 56.28% | 7728 |
| | Con-AM | SS-Con | 9971 | 5571 | 55.87% | 7705 |
| | ConCen-AM | SS-ConCen | 9868 | 5555 | 56.29% | 7657 |
| Random sampling | Cen-AM | RS-Cen | 9963 | 5615 | 56.36% | 7753 |
| | Con-AM | RS-Con | 10,056 | 5674 | 56.42% | 7959 |
| | ConCen-AM | RS-ConCen | 9927 | 5562 | 56.03% | 7689 |

To ensure the annotation quality, the annotation results need to be proofread repeatedly, and the annotation process is time-consuming and laborious. In addition, the images of historical Tibetan documents have different degrees of quality fading, which increases the difficulty of annotation. Therefore, after the steps of annotation and proofreading, each set of data has approximately 5500 character blocks available, which is about 56% of the total.

We use the trained character recognition model to recognize six sets of character data after shielding syllable points in Table 3 to verify the effect of character segmentation. The character recognition results are shown in Figure 12. According to the distribution of character recognition results, the recognition results of 30 rounds of training are divided into the following four parts: (1) the number of model training rounds is between 1 and 10 rounds, and the overall character recognition effect shows an upward trend; (2) the number of model training rounds is between 11 and 14, and the character recognition effect shows a downward trend; (3) the number of model training rounds shows an upward trend for between 15 and 18 rounds; (4) the number of training rounds of the model tends to be stable for between 19 and 30 rounds. The highest recognition accuracy of the above four parts is shown in Table 4.

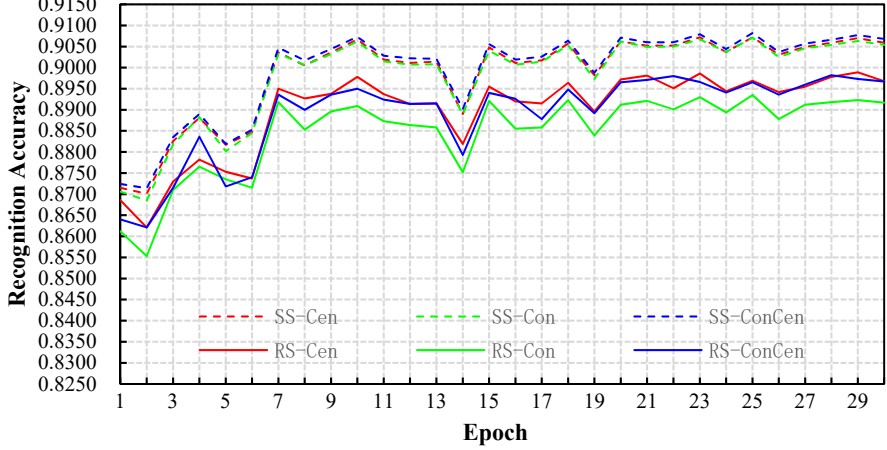

**Figure 12.** Recognition result statistics of correct character segmentation.

**Table 4.** Highest recognition rate statistics.

| Epoch | SS-Cen | SS-Con | SS-ConCen | RS-Cen | RS-Con | RS-ConCen |
|-------|--------|--------|-----------|--------|--------|-----------|
| 1–10 | 0.9067 | 0.9062 | 0.9073 | 0.8978 | 0.8909 | 0.8950 |
| 11–14 | 0.9019 | 0.9015 | 0.9028 | 0.8937 | 0.8873 | 0.8924 |
| 15–18 | 0.9057 | 0.9053 | 0.9064 | 0.8964 | 0.8923 | 0.8948 |
| 19–30 | 0.9072 | 0.9071 | 0.9082 | 0.8989 | 0.8935 | 0.8982 |

*4.4. Limitation Analysis*

Although this work constructs 610 classes of historical Uchen Tibetan document character datasets, which solves the problem of imbalance between the number of classes and the number of samples to a certain extent, in the face of complex historical Tibetan documents, the proposed character dataset construction method has the following certain limitations:

Noise interference. If there is noise in the character block, the interference of noise with the actual number of the characters in the process of extracting characters by using Algorithm 1 will further lead to inconsistency between the extracted character number sequence and the real sequence.

Low-frequency characters. Historical Tibetan documents are difficult to obtain, resulting in a small number of samples of low-frequency character classes and a large number of character samples to be augmented, which reduces the proportion of real samples in the character dataset.

## 5. Conclusions

We propose the construction method of the historical Uchen Tibetan document character dataset, which includes the following four parts: (1) manually annotating character blocks; (2) using the character image extraction method based on real position information; (3) using "Tibetan-Sanskrit" handwritten samples based on component combination to generate ordinary characters and recollect or synthesize special symbols; (4) verifying the effect of the dataset based on the character recognition of historical documents based on a deep neural network. The experimental results show that our method solves the challenge of historical Tibetan document characters in the process of dataset construction to a certain extent, constructs a 610-class character dataset, and has high accuracy in character recognition, which provides support for the follow-up research on historical Tibetan document analysis and recognition.

Although our method is proposed to solve the construction of historical Tibetan document character dataset, it can provide a reference for facing the following similar challenges of historical document character dataset construction: (1) the small scale of document data and (2) the characters being composed of fixed structural units. The proposed method can be applied to the following scenarios: (a) digitization and recognition of historical Tibetan cultural relics and (b) character recognition of other versions of historical Tibetan documents.

**Author Contributions:** Conceptualization, C.Z. and W.W.; methodology, C.Z.; software, C.Z. and G.Z.; validation, C.Z., W.W. and G.Z.; resources, C.Z. and W.W.; writing—original draft preparation, C.Z.; writing—review and editing, C.Z. and W.W.; supervision, W.W. All authors have read and agreed to the published version of the manuscript.

**Funding:** This work was supported by the National Natural Science Foundation of China un-der Grant 61772430 and Grant 62166036, Science and Technology Research Program of Chongqing Education Commission under Grant KJQN202101608, Research Program of Chongqing University of Education under Grant KY202118C.

**Data Availability Statement:** Not applicable.

**Conflicts of Interest:** The authors declare no conflict of interest.

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
