# Peer review of "Construction of a Character Dataset for Historical Uchen Tibetan Documents under Low-Resource Conditions"

_electronics, doi:10.3390/electronics11233919_

Round 1
Reviewer 1 Report
The paper is devoted to construction of a dataset of Uchen Tibetan handwritten characters, training and testing classifier network thereof. The work contains all necessary parts: introduction, related work, methodology, experiments, and conclusion. English language itself has some flaws but this does not affect understanding.
The presented study can be split into two parts: the first is collecting the database and the second is training and testing the classifier. The first part in its turn consists of character annotation, character image extraction, and dataset construction/augmentation.
Annotation is done manually by experts. The question is: were there any contradictions in annotation? I can hardly believe that all eight experts have annotated completely equal. How many contradictions were there and how they were resolved?
Connected question. At string 301 (page 11) it is claimed that 56% of data were used. How these data were selected?
Concerning character image extraction. I cannot understand it. As I can see from line 101 (page 3) and figures, the characters are already outlined in images with different colors (unique color corresponds to a single character) by 'previous work'. To my mind, starting from this point obtaining images of distinct characters and the order (position in image) is a schoolboy's task, which is hardly worth to mention. Just enumerate all unique RGB triplets in the image, find bounding box for pixels with each triplet, and sort the X coordinated of centers of bounding boxes in increasing order, that all. Why do the authors use Algorithm 1 and Algorithm 2 here? Especially for the Algorithm 1: is there any sense in a special (and very poorly described) algorithm of producing 32-bit integer from RGB triplet, while RGB triplet is by its nature already 24-bit integer?
In general, all 'algorithms' are poorly and incompletely described. For instance, where from do we take 'NsingleChar' value in Algorithm 3? Or how do we rotate strokes in Algorithm 4? How do we extract upper and lower strokes? What does it mean 'separate images'? And so on.
The description of algorithms (and the whole first stage) should be rewritten in clear manner by using formulas and formal definitions.
The second part conforms to standard practices of this kind and does not arise obvious questions. Maybe the questions will appear after clarifying the previous part.
Minor drawbacks.
1. Figure 5. As I can see, (b), (c), and (d) are G, B, and R components of (a) rather than R, G, B.
2. Figure 5. (e) is a plain grey square, useless, should be removed.
By its contents the paper is not quite suitable to Electronics MDPI journal. It fits better to one of 'Big data & Cognitive Computing', Heritage, Information, Languages MDPI journals.
Author Response
Reviewer #1 Concern #1: The paper is devoted to construction of a dataset of Uchen Tibetan handwritten characters, training and testing classifier network thereof. The work contains all necessary parts: introduction, related work, methodology, experiments, and conclusion. English language itself has some flaws but this does not affect understanding.
Author action: Thank you for your comments. We have carefully checked the grammar and expression of the manuscript, and revised and improved the English language problems.
Reviewer #1 Concern #2: The presented study can be split into two parts: the first is collecting the database and the second is training and testing the classifier. The first part in its turn consists of character annotation, character image extraction, and dataset construction/augmentation.
Author action: Thank you for your comments. In order to show our work more clearly, we split the main structure of the paper into two parts: character dataset construction and character recognition.
Reviewer #1 Concern #3: Annotation is done manually by experts. The question is: were there any contradictions in annotation? I can hardly believe that all eight experts have annotated completely equal. How many contradictions were there and how they were resolved?
Author action: Thank you for your questions. Because we did not express clearly the contents for annotation in the paper, which caused a little misunderstanding to the reviewers. We have carefully revised the expression of this part: “To ensure the accuracy of character annotation, we choose 8 Tibetans whose mother tongue is Tibetan and who have completed their graduate education as annotators to complete the annotation. Their annotation tasks do not overlap. An expert in the field of Tibetan language studies proofread their annotation results.”
Reviewer #1 Concern #4: Connected question. At string 301 (page 11) it is claimed that 56% of data were used. How these data were selected?
Author action: Thank you for your question. To ensure the annotation quality, the annotation results need to be proofread repeatedly, and the annotation process is time-consuming and laborious. In addition, the images of historical Tibetan documents have different degrees of quality fading, which in-creases the difficulty of annotation. Therefore, after the steps of annotation and proofreading, each set of data has approximately 5500 character blocks available, which is about 56% of the total. The data with valid annotation accounted for 56% of the total, rather than 56% of the selected data.
Reviewer #1 Concern #5: Concerning character image extraction. I cannot understand it. As I can see from line 101 (page 3) and figures, the characters are already outlined in images with different colors (unique color corresponds to a single character) by 'previous work'. To my mind, starting from this point obtaining images of distinct characters and the order (position in image) is a schoolboy's task, which is hardly worth to mention. Just enumerate all unique RGB triplets in the image, find bounding box for pixels with each triplet, and sort the X coordinated of centers of bounding boxes in increasing order, that all. Why do the authors use Algorithm 1 and Algorithm 2 here? Especially for the Algorithm 1: is there any sense in a special (and very poorly described) algorithm of producing 32-bit integer from RGB triplet, while RGB triplet is by its nature already 24-bit integer?
Author action: Thank you for your questions.
(1) If the coloring order is fixed, it is really easy to extract characters in order from character blocks that have completed character segmentation in the previous work. However, the color values of characters between each character block in this work are not in a fixed order, character strokes are a certain degree of arbitrariness, and the character strokes are offset from left to right. Therefore, it is difficult to extract the characters in sequence by using the centroid or coordinate position information of the connected component, which easily leads to disorder between the extracted character numbering sequence and the sequence. To solve the above problems, we first realize color value conversion by performing secondary encoding on the color values of the RGB three channels into a 32-bit grayscale and then extract the single characters, which can avoid the possibility that the color values cannot be uniquely corresponding when the RGB image is converted to the gray-scale image.
(2) Algorithm 1 is simple to implement, so we have removed it.
Reviewer #1 Concern #6: In general, all 'algorithms' are poorly and incompletely described. For instance, where from do we take 'NsingleChar' value in Algorithm 3? Or how do we rotate strokes in Algorithm 4? How do we extract upper and lower strokes? What does it mean 'separate images'? And so on.
Author action: Thank you for your questions.
(1) We have revised three algorithms into clearer pseudo code form.
(2) In the previous version, 'NsingleChar' is the number of characters after extraction.
(3) The specific operation of rotating strokes is to rotate the stroke 90 degrees counterclockwise.
(4) The extraction of upper and lower strokes is completed by using the Y coordinate of the centroid. Frist, the connected component analysis of strokes is carried out. Second, upper and lower strokes are determined according to the size relationship of the Y coordinate of the centroid. Final, the upper and lower strokes are extracted.
(5) In the previous version, 'separate images' refer to the upper and lower strokes after extraction, and we have revised this expression.
Reviewer #1 Concern #7: The description of algorithms (and the whole first stage) should be rewritten in clear manner by using formulas and formal definitions.
Author action: Thank you for your comments. We have revised three algorithms into pseudo code.
|
Algorithm 1: Character extraction algorithm based on real location information |
|
Input: Encoded character block ECB Output: Single character SC 1: H, W ← size function(ECB) 2: NC ← count the number of colors in ECB 3: for l = 1: 2: H do 4: ECBL ← ECB(1: l,:) = 0, ECB(l+2, :) = 0 5: NCL ← count the number of colors in ECBL 6: if NC == NCL 7: for c = 1: NC do 8: for h, w = 1: H, 1: W do 9: if ECB(h, w) ≠ NCL(c) 10: ECB(h, w) = 0 11: end if 12: end for 13: SC ← cropping function(ECB) 14: end for 15: end if 16: end for |
|
Algorithm 2: Construction of the algorithm of the character dataset based on real position sequence matching |
|
Input: Single character annotations SCA, Single characters SC Output: Preliminary character dataset PCD 1: NSCA ← count the number of SCA 2: for d = 1 : NSCA do 3: MN ← read the mapping name of SCA[d] 4: ISEF ← exist the class folder of SCA[d] or not 5: if not ISEF 6: CF ← create the class folder of SCA[d] 7: the MN in SC is written into CF //the MN corresponds to a single character name 8: else 9: the MN in SC is written into CF 10: end if 11: PCD ← save function(CF) //each CF represents a character class 12: end for |
|
Algorithm 3: Special character synthesis algorithm based on the random distance between upper and lower strokes |
|
Input: Special symbols SS Output: Synthesized special symbols SSS 1: NSS ← count the number of SS 2: for s = 1 : NSS do 3: SSU, SSL ← extract the upper and lower strokes of SS //use the Y coordinate of centroid 4: SSUs, SSLs ← rotate function(SSU, SSL) //rotate 90 degrees counterclockwise 5: RI ← generate a random integer in [-2,12] //[-2,12] is the priori distance range 6: SSL, SSU ← SSUs, SSLs //the upper(lower) stroke is as the lower(upper) stroke of SSS 7: SSS ← synthesis function(SSL, SSU, RI) 8: end for |
Reviewer #1 Concern #8: The second part conforms to standard practices of this kind and does not arise obvious questions. Maybe the questions will appear after clarifying the previous part.
Author action: Thank you for your suggestions. We have revised the first part (dataset construction) and improved the second part (character recognition).
Reviewer #1 Concern #9: Minor drawbacks. 1). Figure 5. As I can see, (b), (c), and (d) are G, B, and R components of (a) rather than R, G, B. 2). Figure 5. (e) is a plain grey square, useless, should be removed.
Author action: Thank you for your suggestions. We carefully checked the content involved in Figure 5 and found that it was not a necessary part. We have removed Figure 5.

Reviewer 2 Report
1) The main reason for adopting the proposed method is missing in the abstract.
2) Most of the literature papers are clubbed together in the introduction section. It failed to provide a critical analysis on other works.
3) How many images are used? Are all the alphabets of Tibetan language is encountered in this research work?
4) How do you quantify the word "low" in low resource conditions?
5) I find many direction-based features in most of the alphabets of this language. If you do augmentation, say rotation, don't you think the meaning of the alphabet is changed and leads to wrong detection? Justify how augmentation is useful for your data
6) What methods are used for extraction and recognition?
7) How do you assign code for each character?
8) Does the presence of color of the same character leads to wrong detection?
9) What is the architecture used in the deep learning model? More information must be given on that.
10) How do you perform sampling? Is there any loss of accuracy because of sampling?
Author Response
Reviewer #2 Concern #1: The main reason for adopting the proposed method is missing in the abstract.
Author action: Thank you for your comments. We think this is an excellent suggestion and have carefully revised the abstract. The added contents are as follows.
“The results of character segmentation research in the previous work are presented by coloring the characters with different color values. On this basis, the characters are annotated, and the character images corresponding to the annotation are extracted to construct a character dataset.”
Reviewer #2 Concern #2: Most of the literature papers are clubbed together in the introduction section. It failed to provide a critical analysis on other works.
Author action: Thank you for your comments. We have carefully revised the part concerning literature compression. We have carefully revised the content of the literature merging part. All the literatures cited in this paper have important reference value for this work, especially the document image binarization, layout analysis, text line segmentation and character segmentation, which have laid an important foundation for this work. The revised contents are as follows.
“Since 1991, Kojima et al. began to study the analysis and recognition research on woodcut Tibetan documents, including character recognition [1,2], feature extraction [3] and other works. However, these works did not involve the construction of character datasets. More than a decade later, some researchers conducted relevant research in layout analysis [4], text line segmentation [5], character segmentation [6,7] on different versions of historical Tibetan document data. Since 2017, Our research group have conducted more analysis and recognition research on historical Tibetan document images. Han et al. proposed a binarization approach based on several image processing steps, which achieved high performance in image binarization [8]. Zhao et al. proposed an attention U-Net-based binarization approach for the historical Tibetan document images [9]. Zhou et al., Wang et al. and Hu et al. proposed text line segmentation method based on contour curve tracking [10], based on the connected component Analysis [11] and combined local baseline and connected component [12] for Tibetan historical documents respectively. In addition, in the aspect of layout analysis, Zhao et al. proposed accurate fine-grained layout analysis for the historical Tibetan document based on the instance segmentation [13]. Zhang et al. studied char-acter segmentation on the basis of previous work, which provides data for the construction of character dataset in this work [14].”
Reviewer #2 Concern #3: How many images are used? Are all the alphabets of Tibetan language is encountered in this research work?
Author action: Thank you for your questions.
(1) We used the 212 historical Tibetan document images, the size of each image is about 5500 × 1300 pixels. The data used in this paper is from the results of character segmentation in the previous work [14], so we did not indicate the number of images.
(2) Modern Tibetan is composed of 30 consonants and 4 vowels. Ancient Tibetan also contains a number of Sanskrit Tibetan, consisting of 30 consonants, 5 reverse alphabets, and the upper vowel symbols. Therefore, the number of Sanskrit Tibetan is more than that of modern Tibetan. The character dataset constructed of historical Uchen Tibetan documents in this paper almost covers modern Tibetan and a small amount of Sanskrit Tibetan.
Reviewer #2 Concern #4: How do you quantify the word "low" in low resource conditions?
Author action: Thank you for your question. The word “low” in low resource conditions has different definition criteria for different tasks. The data used in this paper comes from the results of character segmentation in the previous work, and the data available for character segmentation is limited. Therefore, the data in this paper belongs to low resources.
Reviewer #2 Concern #5: I find many direction-based features in most of the alphabets of this language. If you do augmentation, say rotation, don't you think the meaning of the alphabet is changed and leads to wrong detection? Justify how augmentation is useful for your data.
Author action: Thank you for your questions. For characters with directional characteristics, detection and recognition errors may occur after rotation. In this paper, the rotation method is used to achieve the augmentation only for the classes of characters whose upper and lower strokes are completely symmetrical, such as character “”. Rotation can increase the diversity of characters without changing their features.
Reviewer #2 Concern #6: What methods are used for extraction and recognition?
Author action: Thank you for your question. The methods of character extraction and character recognition are as follows.
(1) Character extraction: The characters of historical Uchen Tibetan documents belong are handwritten, and their stroke positions and shapes are quite different. In addition, the interference of three channel images in character extraction makes it difficult to extract the characters in sequence by using the centroid or coordinate position information of the connected component, which easily leads to disorder between the extracted character numbering sequence and the sequence. To solve the above problems, we first realize color value conversion by performing secondary encoding on the color values of the RGB three channels into a 32-bit grayscale and then extract the single characters, which can avoid the possibility that the color values cannot be uniquely corresponding when the RGB image is converted to the gray-scale image. We extract characters from encoded character blocks based on real location information (Algorithm 1).
|
Algorithm 1: Character extraction algorithm based on real location information |
|
Input: Encoded character block ECB Output: Single character SC 1: H, W ← size function(ECB) 2: NC ← count the number of colors in ECB 3: for l = 1: 2: H do 4: ECBL ← ECB(1: l,:) = 0, ECB(l+2, :) = 0 5: NCL ← count the number of colors in ECBL 6: if NC == NCL 7: for c = 1: NC do 8: for h, w = 1: H, 1: W do 9: if ECB(h, w) ≠ NCL(c) 10: ECB(h, w) = 0 11: end if 12: end for 13: SC ← cropping function(ECB) 14: end for 15: end if 16: end for |
(2) Character recognition: The main purpose of character recognition in this paper is to verify the quality of the con-structed character dataset. Considering that all the samples in the character dataset are single characters, the classical convolutional neural network LeNet-5 is used in the character recognition model (Figure 5). The character dataset comes from the constructed historical Uchen Tibetan character dataset, which basically covers 610 classes of the most commonly used characters and symbols in historical Uchen Tibetan documents, with 700 image samples in each class. The character dataset is divided into a training subset and verification subset at a ratio of 8:2, and the test data come from the correct characters obtained by character segmentation.
Figure 5. Character recognition model framework.
Reviewer #2 Concern #7: How do you assign code for each character?
Author action: Thank you for your question. Tibetan characters is composed of consonants and vowels and symbols according to the grammar rules (Figure 3). Each alphabet and symbol correspond to a Unicode (see reference [18]: Tibetan (0F00-0FFF), The Unicode Standard, Version 14.0).
Reviewer #2 Concern #8: Does the presence of color of the same character leads to wrong detection?
Author action: Thank you for your question. In the previous phase of character segmentation, different characters are rendered with different colors to distinguish them from the rest of the characters. Even if there are the same characters in a character block, their colors are not the same. Even if there are the same characters in a character block, their colors are different. Therefore, color of the same character does not lead to wrong detection.
Reviewer #2 Concern #9: What is the architecture used in the deep learning model? More information must be given on that.
Author action: Thank you for your comment. We have added the framework of the deep learning model and other relevant information. The added contents are as follows.
“The framework used for model training is Pytorch, and the computing platform is CPU (Core i7-9700 3.00 GHz). The network model is trained using the CrossEntropyLoss loss function and the SGD optimizer (learning rate=0.001, momentum=0.9). Batch size and training epoch are set to 64 and 30 respectively.”
Reviewer #2 Concern #10: How do you perform sampling? Is there any loss of accuracy because of sampling?
Author action: Thank you for your questions.
(1) The data to be recognized comes from 6 sets of correctly segmented character blocks, which are obtained through systematic sampling and random sampling (Table 3). The systematic sampling is sampled every 10 character blocks to ensure every text line can be covered. In order to ensure the validity of sampling results, we also adopted 10% random sampling.
(2) There is almost no loss of accuracy in selecting data by sampling. Because, there is little difference between the statistical results of systematic sampling and random sampling.

Round 2
Reviewer 1 Report
The authors have updated the paper according to the comments. It is suitable for publication now.
Reviewer 2 Report
It can be accepted now